# Biologic Impact of Different Ultra-Low-Fluence Irradiations in Human Fibroblasts

**DOI:** 10.3390/life10080154

**Published:** 2020-08-18

**Authors:** Masao Suzuki, Yukio Uchihori, Hisashi Kitamura, Masakazu Oikawa, Teruaki Konishi

**Affiliations:** 1Department of Basic Medical Sciences for Radiation Damages, National Institute of Radiological Sciences, National Institutes for Quantum and Radiological Science and Technology, 4-9-1 Anagawa, Chiba 263-8555, Japan; 2Department of Research Planning and Promotion, Quantum Medical Science Directorate, National Institutes for Quantum and Radiological Science and Technology, 4-9-1 Anagawa, Chiba 263-8555, Japan; uchihori.yukio@qst.go.jp; 3Department of Radiation Emergency Management, Center for Advanced Radiation Emergency Medicine, National Institutes for Quantum and Radiological Science and Technology, 4-9-1 Anagawa, Chiba 263-8555, Japan; kitamura.hisashi@qst.go.jp; 4Department of Accelerator and Medical Physics, National Institute of Radiological Sciences, National Institutes for Quantum and Radiological Science and Technology, 4-9-1 Anagawa, Chiba 263-8555, Japan; oikawa.masakazu@qst.go.jp; 5Single Cell Radiation Biology Group, Institute for Quantum Life Science, National Institutes for Quantum and Radiological Science and Technology, 4-9-1 Anagawa, Chiba 263-8555, Japan; konishi.teruaki@qst.go.jp

**Keywords:** ultra-low-fluence irradiation, different radiation types, linear-energy transfer, cell death, gene mutation, bystander effect, radio-adaptive response, genomic instability, gap-junction mediated cell-to-cell communication

## Abstract

In this study, we aimed to evaluate the cellular response of healthy human fibroblasts induced by different types of ultra-low-fluence radiations, including gamma rays, neutrons and high linear energy transfer (LET) heavy ions. NB1RGB cells were pretreated with ultra-low-fluence radiations (~0.1 cGy/7–8 h) of ^137^Cs gamma rays, ^241^Am–Be neutrons, helium, carbon and iron ions before being exposed to an X-ray-challenging dose (1.5 Gy). Helium (LET = 2.3 keV/µm), carbon (LET = 13.3 keV/µm) and iron (LET = 200 keV/µm) ions were generated with the Heavy Ion Medical Accelerator in Chiba (HIMAC), Japan. No differences in cell death—measured by colony-forming assay—were observed regardless of the radiation type applied. In contrast, mutation frequency, which was detected through cell transformation into 6-thioguanine resistant clones, was 1.9 and 4.0 times higher in cells pretreated with helium and carbon ions, respectively, compared to cells exposed to X-ray-challenging dose alone. Moreover, cells pretreated with iron ions or gamma-rays showed a mutation frequency similar to cells exposed to X-ray-challenging dose alone, while cells pretreated with neutrons had 0.15 times less mutations. These results show that cellular responses triggered by ultra-low-fluence irradiations are radiation-quality dependent. Altogether, this study shows that ultra-low-fluence irradiations with the same level as those reported in the International Space Station are capable of inducing different cellular responses, including radio-adaptive responses triggered by neutrons and genomic instability mediated by high-LET heavy ions, while electromagnetic radiations (gamma rays) seem to have no biologic impact.

## 1. Introduction

The risk of cancer after exposure to high and moderate doses of radiation is relatively well known and is mostly based on detailed epidemiological data collected from the Japanese atomic bomb survivors in Hiroshima and Nagasaki [1]. However, risks associated with low radiation doses are generally extrapolated from available high-dose data and experimental studies that commonly use acute, high-dose, high-dose-rate radiation exposure. Additionally, most studies designed to evaluate radiation risks only take into consideration the radiobiological effects on cells or tissues actually irradiated. Studies of biologic responses induced by different radiation types, such as electromagnetic and particle radiation, can provide very important information to assess the human risk of radiation exposure in aircrafts and spacecrafts, as well as the risk of exposure to environmental and medical radiation. In particular, the study of biologic responses to low-dose rate or low-fluence irradiations with similar exposure levels as reported in the International Space Station (ISS) or a spacecraft (~0.05–0.1 cGy/day) can provide important clues on the outcome of the Fukushima Daiichi Nuclear Power Plant accident.

Radiation-induced genomic instability can be observed in cells at delayed periods after irradiation. Compared with the direct effects of radiation, which are induced by direct energy deposition in cells, radiation-induced genomic instability may include non-targeted effects that are often seen in cells not subjected to direct radiation incidence. In such cases, the radiation may hit cells through indirect communication between the radiation-targeted cells with non-irradiated neighboring cells, leading to radiobiological bystander effects. In vitro studies have demonstrated that radiation-induced instability can be determined by several cellular effects, including chromosomal damage and changes in gene expressions [2,3,4,5,6,7,8]. In a previous study we have showed that in vitro lifespan of healthy human fibroblasts exposed to very low dose of mixed radiations (0.14 cGy/day), such as gamma rays, neutrons and heavy ions, was shortened compared with non-irradiated control cells [9]. However, the underlying cellular and molecular mechanisms triggered by low-dose/rate radiations remain unclear, particularly those related to high linear energy transfer (LET) radiations. A better understanding of radiation-induced biologic instability may have important implications for risk assessment of low-dose/rate terrestrial and cosmic radiation, as well as in accidents such as that of Fukushima Daiichi Nuclear Power Plant in 2011.

Cells can undergo a process named ‘radio-adaptive response’, in which cells irradiated with a sublethal dose of ionizing radiation (adaptive dose or priming dose) become less sensitive to subsequent irradiation with higher doses (challenging dose). This radio-adaptive response was first described as reduced frequency of X-ray-induced chromosomal aberrations in stimulated human lymphocytes co-cultured with radioactive ^3^H-thymidine [10]. Subsequently, radio-adaptive responses have been reported in several biologic settings [11,12,13,14,15,16,17,18]. A similar adaptive response has also been reported in clinical settings, in patients undergoing treatments for thyroid diseases [19]. Vares et al. also demonstrated that 0.01 Gy of high-LET heavy-ion beams, such as carbon and neon ions, used as priming irradiations could trigger adaptive cellular responses in vitro for challenging doses of 1–4 Gy of high LET heavy-ion beams [20]. Nevertheless, available studies on radio-adaptive responses have presented conflicting results, which could be due to experimental factors. In this study, we evaluated the radiation-quality dependence in cellular responses induced by ultra-low-fluence irradiations, including gamma rays, neutrons and heavy ions, at levels similar to that reported in the ISS and other spacecrafts.

Radiation-induced bystander effects are described as the ability of cells affected by irradiation to convey manifestations of damage to neighbor cells that are not directly irradiated, and the effects have resulted from some types of communication or signaling between direct irradiated and non-irradiated cells. The first report for the bystander cellular effect by Nagasawa and Little was the study that very low doses of alpha-particle irradiation induced sister chromatid exchanges in >30% of Chinese hamster ovary (CHO) cells, even though less than 1% of the cell nuclei in the cell population was estimated to be traversed by 3.7-MeV alpha particles from ^238^Pu [21]. In our recent study, high-LET carbon-ion beams (LET~309 keV/µm) could induce lethal and mutagenic bystander effects in healthy human fibroblasts and show that gap-junction mediated cell-to-cell communication played an important role for inducing the bystander effects using the specific inhibitor of gap junctions (40 µM of gamma-hexachlorocyclohexane) [22].

## 2. Materials and Methods

### 2.1. Human Cells and Culture Conditions

Normal human skin fibroblasts (NB1RGB cells, Cell No. RCB0222, RIKEN BioResource Center Cell Bank) were used [22]. Cell culture was performed in Eagle’s minimum essential medium (MEM) included kanamycin (60 mg/L) and supplemented with 10% fetal bovine serum (HyClone KTE31760, HyClone Laboratories, Inc., Logan, UT, USA) in a 5%-CO_2_ incubator at 37 °C. Frozen cell stocks, which were kept in liquid nitrogen at passage 6 (total population doubling number = 16.4), were thawed and kept in normal culture conditions until the cells reached a confluent state. Two days before irradiation, when the cells were confluent and about 93% were in G_1_- or G_0_-phase as determined by flow cytometry (data not shown), the cells were trypsinized and seeded (passage 8) in 25 cm^2^ culture flasks (cell culture flask, Falcon 353014, Corning, Incorporated, Corning, NY, USA) at a density of ${8\;\times \;10}^{5}$ cells per flask. The doubling-time of the cells was around 24 h and the seeding efficiency was over 40% at passage 8 for cells to be used on colony-forming assays.

### 2.2. Pretreatment with Ultra-Low-Fluence Irradiation

NB1RGB cells were pretreated with ultra-low-fluence irradiations (~0.1 cGy/7–8 h) of ^137^Cs gamma rays, neutrons, helium ions (150 MeV/n, LET = 2.3 keV/µm), carbon ions (290 MeV/n, LET = 13.3 keV/µm) and iron ions (500 MeV/n, LET = 200 keV/µm), before undergoing irradiation with 200 kV X-ray-challenging dose (1.5 Gy) filtered with 0.5-mm Al and 0.5-mm Cu at 0.98 Gy/min. Pretreatment using ultra-low-fluence neutrons was carried out with a ^241^Am–Be neutron source (maximum energy: 11.5 MeV, average energy: 5.0 MeV). Contamination of gamma rays was estimated to be around 15% of the total dose at the sample position. Heavy ions were produced with the Heavy Ion Medical Accelerator in Chiba (HIMAC) at the National Institute for Quantum and Radiological Science and Technology (QST) in Japan. The pretreatment protocol with ultra-low-fluence heavy ions was performed using the faint beam mode, which was ~1/1000 of the intensity commonly used in a normal biologic irradiation experiment. The fluence of each ion was counted using a scintillation counter made in polyvinyl toluene (EJ-212, ELJEN Technology, Sweetwater, TX, USA) and they were converted to dose using the formula.
$$\mathrm{Dose}\;\left(\mathrm{Gy}\right)\;=\;\mathrm{Fluence}\left(\mathrm{ions}/{\mathrm{cm}}^{2}\right)\;\times \;\mathrm{LET}\left(\mathrm{keV}/\mu m\right)\;\times \;{1.602\;\times \;10}^{-9}$$

Characteristics of pretreatment of heavy-ion beams used are shown in Table 1. All irradiation experiments were carried out at room temperature (22–24 °C).

### 2.3. Proton Microbeam Irradiation

Proton microbeams (3.4 MeV) were produced by single particle irradiation system to cell (SPICE) in QST [23]. NB1RGB cells were seeded in special microbeam dishes (Figure 1) two days before the microbeam irradiation assay. These dishes were made of stainless steel, had 31 mm in diameter and had a 2.5 µm Mylar polyester film (CAT. No. 100, Chemplex Industries, Inc., Palm City, FL, USA) attached to the bottom of the ring. In order to block the cell-to-cell communications, half of the prepared sample dishes were treated with 40 µM of gamma-hexachlorocyclohexane (Sigma-Aldrich Japan), a specific inhibitor of gap-junction mediated cell-to-cell communication, 24 h before the irradiation. The irradiation protocol was carried out using the 5625 (75 × 75)-cross-stripe irradiation method (Figure 1). The spot size of the microbeam was 10 µm  ×  10 µm and irradiation spots line up 5625 (75  ×  75) neighboring spots at the intervals of 300 µm in all directions. In this experimental setting, the total cell number in the confluent state was measured to be ${4\;\times \;10}^{5}$ cells/irradiation dish. Based on this measured cell density, the rough estimated percentage of cells irradiated with a single proton, under a 10 µm × 10 µm microbeam size, was ${5625\;/\;4\;\times \;10}^{5}\;=\;0.014\;(1.4\%)$. The beam interval of the applied irradiation method was 300 µm, which is considerably bigger than the size of a single cell that was determined to be 21 µm in diameter of cell nuclei under the microscope.

### 2.4. Cell Death Assessment

Within 30 min after starting the above described pretreatment protocol, we irradiated the seeded cells at room temperature with 1.5 Gy of 200 kV X-ray as challenging irradiation dose. Cell death was then determined by colony forming assay. After exposure to X-ray challenging irradiation, different numbers of cells were immediately seeded onto 100-mm culture dishes (Tissue Culture Dish, Falcon 353,003, Corning Incorporated, Corning, NY, USA) to form 60–70 colonies/dish. The colonies were fixed and stained with 20% methanol and 0.2% crystal violet after a 14-day incubation period at 5% CO_2_, 37 °C incubator. Any colony consisting of more than 50 cells was deemed a surviving clone.

### 2.5. Analysis of Mutation Frequency

The potential of X-ray-challenging dose irradiation to induce genetic mutations was evaluated based on the proportion of 6-thioguanine (6TG: CAT. No. 203–03771, Wako Junyaku, Japan)-resistant clones at the *HPRT* (hypoxanthine–guanine phosphoribosyl transferase) locus, which has been described elsewhere [22,24,25]. Briefly, NB1RGB cells were irradiated with X-ray-challenging dose and subsequently cultured in 75 cm^2^ flasks at a density of 1.5 × 10^6^ cells per flask. As these cells reached 6 to 8 population doubling numbers, which is considered enough to allow expression of the mutation, ${1\;\times \;10}^{6}$ cells were plated in 100-mm culture dishes containing MEM supplemented with 40 µM of 6TG. The cultures were maintained for 14 days at 5% CO_2_ and 37 °C incubator and were subsequently fixed with 20% methanol and stained with 0.2% crystal violet. Any colony consisting of more than 50 cells was scored as a 6TG-resistant mutant clone. The mutation frequency was determined as the number of 6TG-resistant colonies per 10^6^ survivors. In both cell death and mutation induction experiments, a specific inhibitor of gap-junctions (40 µM of gamma-hexachlorocyclohexane) was added to the culture medium 24 h before pretreatment with ultra-low-fluence irradiations to the end of irradiations of X-ray-challenging dose to examine the effects of bystander responses via gap-junction mediated cell-to-cell communication. In this experimental condition, no change was observed in the shape of the fibroblasts under a light microscope.

### 2.6. Statistical Analysis

All data for cell death and mutation incidence induced by X-ray-challenging irradiation were calculated from seven independent experiments (independent beam times). Student’s *t*-test was used to identify significant differences between control (X-ray-challenging dose alone) and pretreated irradiation groups, with *p*-value of 0.05 or less representing statistical significance.

## 3. Results

No significant differences were observed in the ability of X-ray-challenging dose to induce death of NB1RGB cells pretreated with different radiation types (Figure 2, Table 2). Similarly, no differences were seen in mutation frequency between cells irradiated with 1.5-Gy X-ray-challenging dose alone or with pretreatment with gamma-ray or iron-ion radiation (Figure 3, Table 3). However, in cells pretreated with high-LET ions, the helium-ion pretreated cells and carbon–ion pretreated cells were found to have ~1.9 and 4 times more mutations, respectively than the control sample with X-ray-challenging dose alone. In contrast, mutation frequency was reduced in fibroblasts pretreated with neutrons, looking like radio-adaptive response (Figure 3). Overall, these results provide evidence that mutation induction potential of low-fluence irradiation depends on the radiation type.

The cells were pretreated with the same fluence (dose) as 0.1 cGy/7–8 h for the different radiation types in the above-mentioned experiments. Then, cell death and mutation frequency were examined using the same-ion hits (15% of cells were hit directly) of helium, carbon and iron ions in the pretreatment protocol. The results showed the same trend as the previous experiments, with no significant differences in cell death (Figure 4) and higher mutation frequency in cells pretreated with helium and carbon ions compared to X-ray-challenging dose alone (Figure 5). These results clearly demonstrate that higher-LET iron ions cannot induce genomic instability, even in cases of higher ion fluence.

To understand the observed phenomena for altered mutation frequency in cells pretreated with neutron, helium and carbon ions, the cells were treated with 40 µM of gamma-hexachlorocyclohexane from 24 h before pretreatment with either neutrons or heavy ions to the end of irradiations with X-ray-challenging dose, focusing bystander effect induced by gap-junction mediated cell-to-cell communication. Results showed to be prevented by the presence of the gap-junction inhibitor, reaching mutation levels similar to X-ray-challenging dose alone (Figure 6). These results provide clear evidence that the observed cellular responses to low-fluence-neutron-induced radio-adaptive response and helium- or carbon-ion-induced genomic instability can be explained by bystander effects related to gap-junction mediated cell-to-cell signals from hit cells.

The mutation induced by 1.5 Gy of the X-ray-challenging dose was clearly reduced in the cells pretreated with neutron irradiation. In the case of neutron irradiation using a ^241^Am–Be neutron source, maximum energy of incident neutrons is 11.5 MeV and average energy is 5.0 MeV and so initial interactions between neutron and cell bear recoiled protons produced from incident neutrons. Next cell death and mutation frequency induced by low-fluence-proton pretreatment were examined using SPICE proton microbeams in order to confirm the biological effects underlying the proton-induced radio-adaptive response. The results showed that X-ray induced mutation frequency was suppressed in cells pretreated with proton microbeams (Figure 7), while no effect was observed in clonogenic cell survival (Figure 8). Moreover, mutation frequency returned to control levels when gap-junction inhibitor was present (Figure 7). Overall, these results suggest that neutron-induced adaptive response is caused by recoiled protons and that gap-junction mediated bystander effect plays an important role to induce such cellular response.

## 4. Discussion

In this study, we chose 0.1 cGy/7–8 h as the standard priming protocol with ultra-low-fluence irradiation applied before 1.5-Gy X-ray-challenging dose, as the absorbed dose for a stay in the ISS was estimated to be 0.05–0.1 cSv/day. According to the Poisson distribution analysis, the percent of cells directly hit by heavy ions of different LET values during pretreatment protocol was estimated to be 61% for helium ions (LET = 2.3 keV/µm), 15% for carbon ions (LET = 13.3 keV/µm) and 1.1% for iron ions (LET = 200 keV/µm) shown in Table 1. These different LET values indicate that most of the cells were not hit directly with the iron-ion beams. The relatively low hit effectiveness could in part explain the differences seen in mutation induction potential between different irradiation types (Figure 3). Additionally, the reported higher frequency of mutations induced by X-ray in fibroblast pretreated with either helium or carbon ions could be due to bystander effect in non-hit cells triggered by genomic instability of hit cells (Figure 6). To better understand the potential mechanisms involved in mutational potential of each radiation type, we next conducted new experiments using the same fluence hit (15% of cells were hit directly) of helium, carbon and iron ions in the pretreatment protocol. The results clearly showed the same trend as the same fluence (dose) experiments of 0.1 cGy/7–8 h, with no significant differences in cell death (Figure 4) and higher mutation frequency in cells pretreated with helium and carbon ions compared to X-ray-challenging dose alone (Figure 5). These results clearly demonstrate that higher-LET iron ions cannot induce genomic instability, even in cases of higher ion fluence.

Next, we further evaluated potential mechanisms for altered mutation frequency in cells pretreated with neutron, helium and carbon ions. We focused on the possibility of the bystander effect induced by gap-junction mediated cell-to-cell communication and examined the mutation potential in the presence of a specific gap-junction inhibitor. The cells were treated with 40 µM of gamma-hexachlorocyclohexane before pretreatment with either neutrons or heavy ions. Results showed that the effects previously seen, such as reduced frequency or promotion of X-ray induced mutation in cells pretreated with neutrons or with helium or carbon ions, respectively, were prevented by the presence of the gap-junction inhibitor, reaching mutation levels similar to X-ray-challenging dose alone (Figure 6). These results provide clear evidence that the observed cellular responses to low-fluence-neutron-induced radio-adaptive response and helium- or carbon-ion-induced genomic instability can be explained by bystander effects related to gap-junction mediated cell-to-cell signals from hit cells.

The significantly reduced mutation frequency in cells pretreated with neutrons (Figure 3 and Figure 6) could in part be explained by initial interactions between neutron and cell bear recoiled protons produced from incident neutrons. A hypothesis is that the radio-adaptive response could be induced by recoiled protons emitted as a consequence of the interaction between primary fast neutrons and surroundings near irradiated cells. To confirm the mechanism underlying the proton-induced radio-adaptive response, we examined cellular response in which just 1.4% of the cells were irradiated using SPICE proton microbeams. The results showed that X-ray induced mutation frequency was suppressed in cells pretreated with proton microbeams (Figure 7), while no effect was observed in clonogenic cell survival (Figure 8). These findings are consistent with other previous results of SPICE proton microbeams used in zebrafish [26]. Moreover, mutation frequency returned to control levels when gap-junction inhibitor was present (Figure 7). Overall, these results suggest that neutron-induced adaptive response is caused by recoiled protons, and that gap-junction mediated bystander effect plays an important role to induce such cellular response.

There is clear evidence that ultra-low-fluence irradiations with the same exposure level as that reported in the ISS or other spacecraft are capable of inducing different cellular responses, including radio-adaptive response to neutrons and genomic instability to high-LET heavy ions, ultimately impacting in the life-span of human cells [9]. In contrast, electromagnetic radiations (gamma rays) seem to have no significant cellular impact. Our study suggests that ultra-low-fluence irradiations are not responsible *per se* for inducing such biological effects; however, they are capable of inducing different cellular responses such as genomic instability and radio-adaptive response through continuous stress. Radiation-quality dependence in such epigenetic effects is still unclear, and several questions remain unanswered. For example, why iron ions cannot induce genomic instability or why do different epigenetic effects, such as genomic instability and radio-adaptive response, occur between neutrons and helium/carbon ions remain unknown. More studies exploring in detail the biologic effects of microbeam radiations are warranted. Altogether, our findings indicate that bystander effect induced by gap-junction mediated cell-to-cell communication is one possible modulator of radiation-induced epigenetic effects. However, the mechanism of radiation-induced bystander effects is still unknown, and we need to study in more detail, e.g., what kinds of molecules bear the pathway of gap-junction mediated cell-to-cell communication. In addition, another pathway, such as a secreted factor to culture medium from direct irradiated cells, is a high-powered candidate for inducing bystander effect. Using healthy human cells, we also observed secreted-factor induced bystander effect [27]. To further elucidate this mechanism, researchers should explore which genes and proteins that could be critical player on bystander effects.

## Figures and Tables

**Figure 1 life-10-00154-f001:**
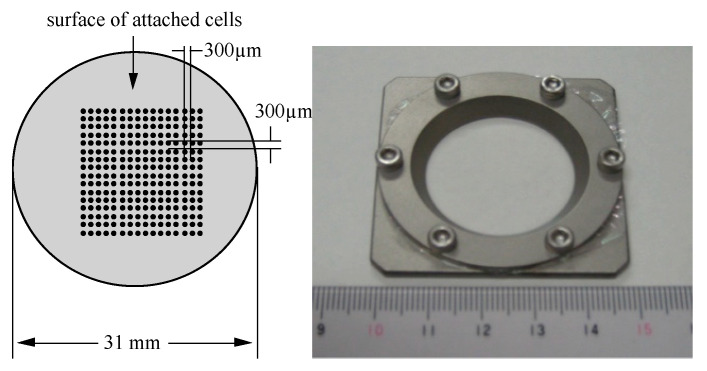
Special dish for proton microbeam irradiation in single particle irradiation system to cell (SPICE). Around 1.4% of total cells were irradiated with a single 3.4-MeV proton before receiving the X-ray-challenging dose under the 5625 (75 × 75)-cross-stripe irradiation method.

**Figure 2 life-10-00154-f002:**
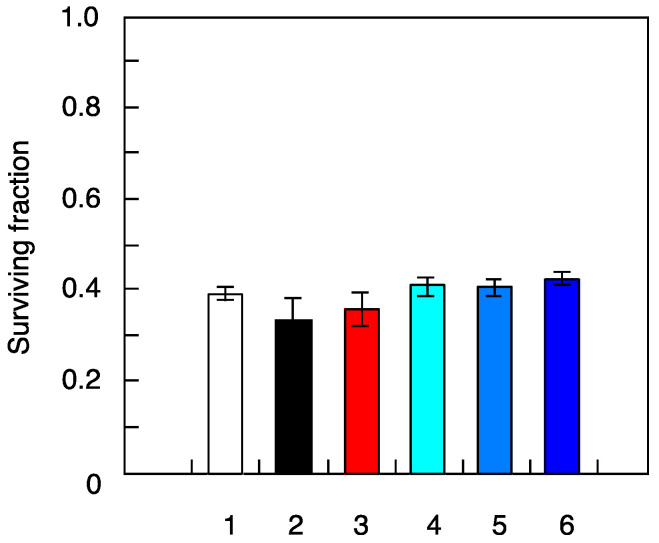
Cell death induced by X-ray-challenging dose in human fibroblasts pretreated with ultra-low-fluence irradiations of different types. NB1RGB cells were pretreated with different types of ultra-low-fluence irradiations followed by 1.5-Gy dose of X-rays, as described in the ‘Material and Methods’ section. The results shown represent the mean and standard errors of 7 independent experiments. 1—X-ray challenge alone, 2—^137^Cs gamma rays pre-treatment plus X-ray challenge, 3—^241^Am–Be neutrons pre-treatment plus X-ray challenge, 4—helium ions pre-treatment plus X-ray challenge, 5—carbon ions pre-treatment plus X-ray challenge, 6—iron ions pre-treatment plus X-ray challenge.

**Figure 3 life-10-00154-f003:**
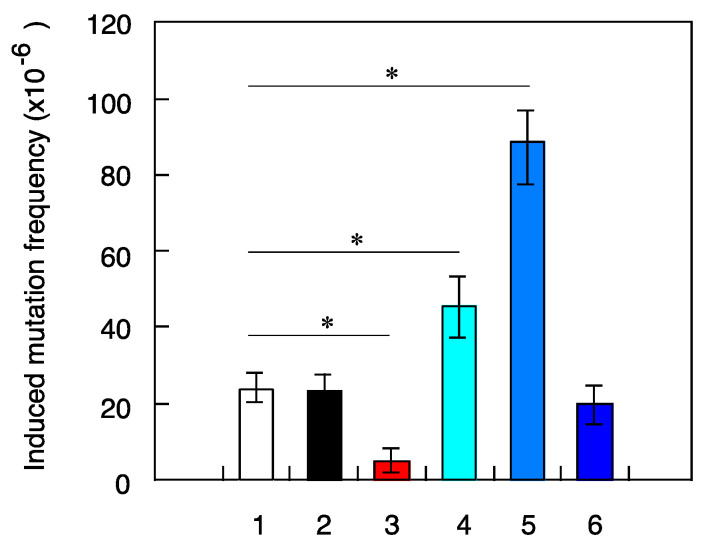
Frequency of mutations induced by X-ray-challenging dose in human fibroblasts pretreated with ultra-low-fluence irradiations of different types. NB1RGB cells were pretreated with different types of ultra-low-fluence irradiations followed by 1.5-Gy dose of X-ray, as described in the ‘Material and Methods’ section. The results shown represent the mean and standard errors of 7 independent experiments and (*) indicates *p* < 0.05. 1—X-ray challenge alone, 2—^137^Cs gamma rays pre-treatment plus X-ray challenge, 3—^241^Am–Be neutrons pre-treatment plus X-ray challenge, 4—helium ions pre-treatment plus X-ray challenge, 5 —carbon ions pre-treatment plus X-ray challenge, 6—iron ions pre-treatment plus X-ray challenge.

**Figure 4 life-10-00154-f004:**
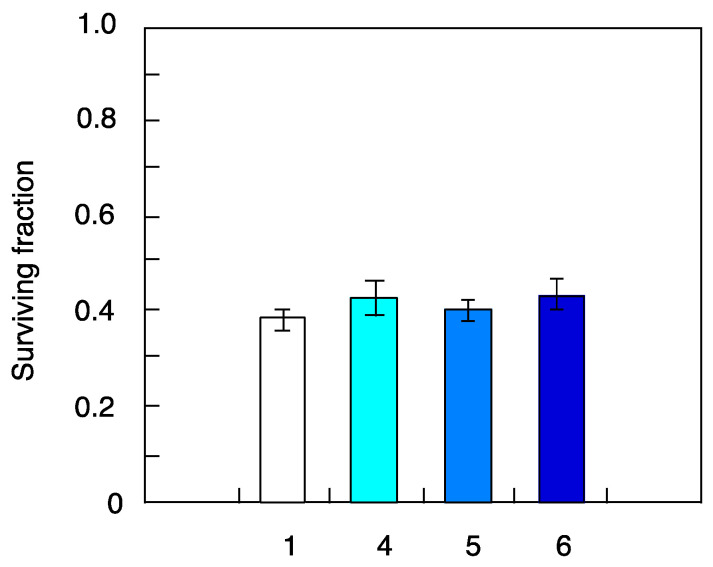
Cell death induced by X-ray-challenging dose in human fibroblasts pretreated with heavy ions with the same fluence hit of 15% of total cells. The results shown represent the mean and standard errors of 7 independent experiments. 1—X-ray challenge alone, 4—helium ions pre-treatment plus X-ray challenge, 5—carbon ions pre-treatment plus X-ray challenge, 6—iron ions pre-treatment plus X-ray challenge.

**Figure 5 life-10-00154-f005:**
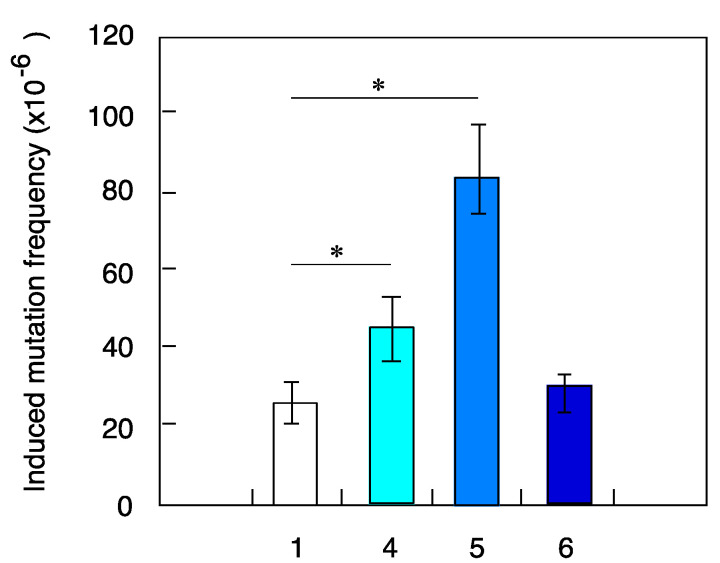
Frequency of mutations induced by X-ray-challenging dose in human fibroblasts pretreated with heavy ions with the same fluence hit of 15% of total cells. The results shown represent the mean and standard errors of 7 independent experiments and (*) indicates *p* < 0.05. 1—X-ray challenge alone, 4—helium ions pretreatment plus X-ray challenge, 5—carbon ions pretreatment plus X-ray challenge, 6—iron ions pretreatment plus X-ray challenge.

**Figure 6 life-10-00154-f006:**
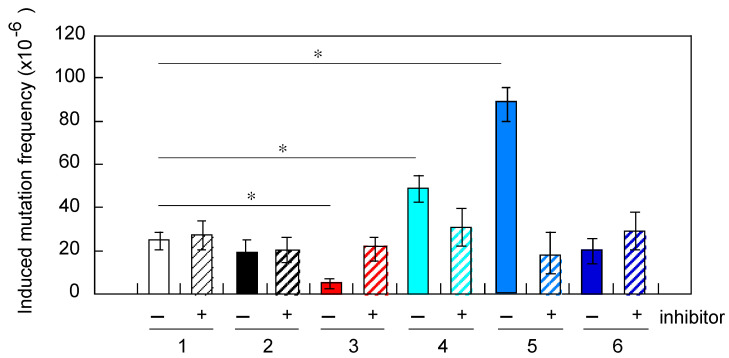
Frequency of mutations induced by X-ray-challenging dose in human fibroblasts pretreated with ultra-low-fluence irradiations of different types. The results shown represent the mean and standard errors of 7 independent experiments and (*) indicates *p* < 0.05. The marks of the horizontal axis are the same as Figure 2, with “+” indicating the presence of a specific inhibitor of gap-junction mediated cell-to-cell communication.

**Figure 7 life-10-00154-f007:**
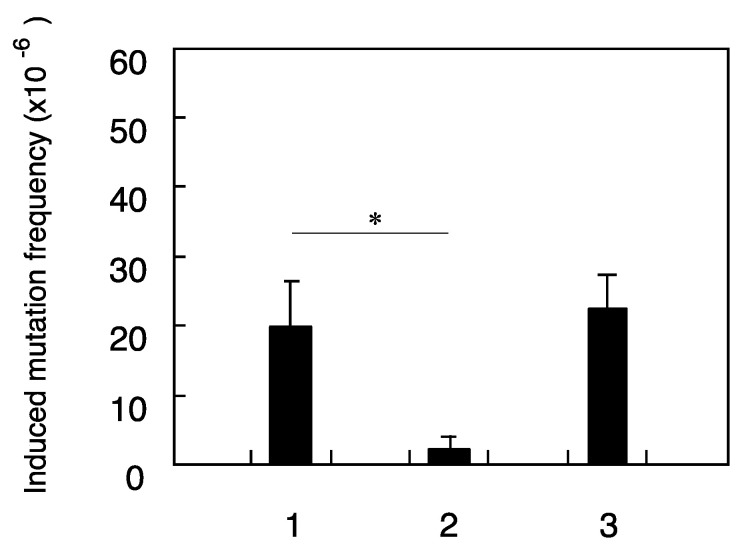
Frequency of mutations induced by X-ray-challenging dose in human fibroblasts pretreated with ultra-low-fluence irradiations of proton microbeams. 1—X-ray challenge alone, 2—single proton pre-treatment plus X-ray challenge, 3—single proton pre-treatment plus X-ray challenge in the presence of gap-junction inhibitor. The results shown represent the mean and standard errors of 7 independent experiments and (*) indicates *p* < 0.05.

**Figure 8 life-10-00154-f008:**
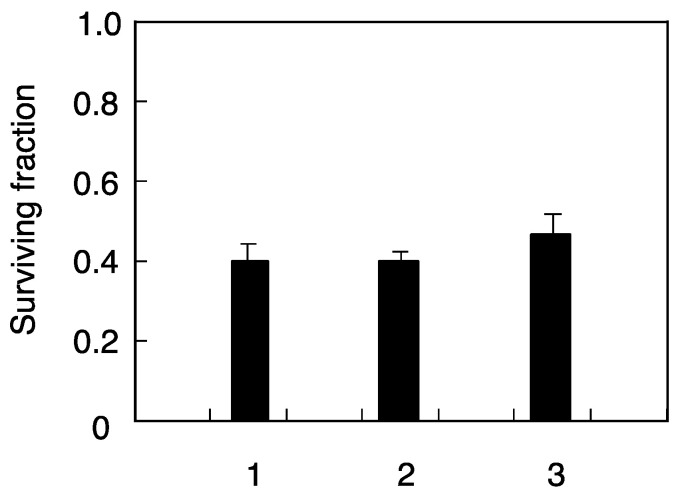
Cell death induced by X-ray-challenging dose in human fibroblasts pretreated with ultra-low-fluence irradiations of proton microbeams. The marks of the horizontal axis are the same with Figure 7. The results shown represent the mean and standard errors of 7 independent experiments.

**Table 1 life-10-00154-t001:** Characteristics of pretreatment of heavy-ion beams

Pretreatment Radiation Types	Energy (MeV/n)	LET (keV/µm)	Fluence at 0.1 cGy (ions/cm^2^)	Average Number of Hits/Nucleus ^a^ (%)
Helium ions	150	2.3	2.71 × 10^5^	61
Carbon ions	290	13.3	4.69 × 10^4^	15
Iron ions	500	200	3.12 × 10^3^	1.1

^a^ Percent of direct hit / cell nucleus were calculated according to the Poisson distribution using the measured diameter of NB1RGB cells (21 µm in diameter).

**Table 2 life-10-00154-t002:** Cell death induced by ultra-low-fluence irradiation in human fibroblasts.

Pretreatment Radiation Types (0.1 cGy/7–8 h)	Cell Survival (%) By X-ray-Challenging Dose (1.5 Gy) ^a^
X-ray-challenging dose 1.5 Gy alone	39.8 ± 1.6
^137^Cs gamma rays	34.8 ± 1.9
^241^Am–Be neutrons	35.7 ± 1.8
Helium ions (LET = 2.3 keV/µm)	41.1 ± 1.7
Carbon ions (LET = 13.3 keV/µm)	40.2 ± 1.8
Iron ions (LET = 200 keV/µm)	42.4 ± 1.3

^a^ Mean and standard errors of 7 independent experiments are shown.

**Table 3 life-10-00154-t003:** Mutation frequency in human fibroblasts pretreated with ultra-low-fluence irradiation.

Pretreatment Radiation Types (0.1 cGy/7–8 h)	Induced Mutation Frequency (×10^−6^) By X-ray-Challenging Dose (1.5 Gy) ^a^
X-ray-challenging dose 1.5 Gy alone	25.7 ± 5.8
^137^Cs gamma rays	24.4 ± 5.6
^241^Am–Be neutrons	6.8 ± 4.0
Helium ions (LET = 2.3 keV/µm)	46.8 ± 6.3
Carbon ions (LET = 13.3 keV/µm)	90.8 ± 7.6
Iron ions (LET = 200 keV/µm)	21.8 ± 5.5

^a^ Mean and standard errors of 7 independent experiments are shown.

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
