# Peer review of "Biologic Impact of Different Ultra-Low-Fluence Irradiations in Human Fibroblasts"

_life, 2020, doi:10.3390/life10080154_

Round 1

Reviewer 1 Report

This manuscript presented systematic assessment of effects by ultra-low-frequency irradiations with a range of sources, and reported qualitatively different impact on cell death and mutations. Highly reproducible experimental system developed by the authors allowed clear conclusions on the main points described in the title and abstract.

Since the effects of low-frequency irradiations which mimic conditions in ISS, this work should have significant value in the field as a basis for discussion on low-orbit radiation effects.

Only concerns from this reviewer is related to descriptions about gap-junction parts:

1. Introduction should include rationale of suspecting contribution of gap-junction in bystander effects. Reference for gap junction inhibitor (and concentration for this cell type, if appropriate) should be included.

2. There could be alternative mechanisms of the bystander effects - discussion should better provide possibilities for other interpretation of the results. Other intracellular events or intercellular communications could mediate such effects. Also, it is possible that cell body shape could be changed by the inhibitor (reduced cell area being exposed by the radiation beam).

Author Response

Our replies for the reviewer’s (1) comments

We appreciate for reviewing our manuscript and insightful comments and valuable suggestions. We revised the manuscript according to the suggestions and comments from the reviewers.

  1. Introduction should include rationale of suspecting contribution of gap-junction in bystander effects. Reference for gap junction inhibitor (and concentration for this cell type, if appropriate) should be included.

We added a description regarding bystander effects, and also referred the role of gap-junction mediated cell-to-cell communication of our recent study using the same human fibroblasts with this paper in the introduction part. We referred one paper in reference as follows;

Radiation-induced bystander effects are described as the ability of cells affected by irradiation to convey manifestations of damage to neighbor cells that are not directly irradiated, and the effects have resulted from some types of communication or signaling between direct irradiated and non-irradiated cells. The first report for the bystander cellular effect by Nagasawa and Littlewas the study that very low doses of alpha-particle irradiation induced sister chromatid exchanges in >30% of Chinese hamster ovary (CHO) cells, even though less than 1% of the cell nuclei in the cell population was estimated to be traversed by 3.7 MeV alpha particles from 238Pu[21]. In our recent study, high-LET carbon-ion beams (LET ~ 309keV/µm) could induce lethal and mutagenic bystander effects inhealthy human fibroblasts and show that gap-junction mediated cell-cell communication played an important role for inducing the bystander effects using the specific inhibitor of gap junctions (40 µM of gamma-hexachlorocyclohexane)[22].

  1. There could be alternative mechanisms of the bystander effects - discussion should better provide possibilities for other interpretation of the results. Other intracellular events or intercellular communications could mediate such effects. Also, it is possible that cell body shape could be changed by the inhibitor (reduced cell area being exposed by the radiation beam).

We added a sentence regarding other pathway for inducing bystander effects referring our paper concerning secreted-factor mediated bystander effect in healthy human cells in the discussion part as follows;

However, the mechanism of radiation-induced bystander effects is still unknown and we need to study in more detail, e.g. what kinds of molecules bear the pathway of gap-junction mediated cell-to-cell communication. Also, other pathway, such as a secreted factor to culture medium from direct irradiated cells, is high-powered candidate for inducing bystander effect. Using healthy human cells we also observed secreted-factor induced bystander effect[27].

And we added regarding the cell body shape by the inhibitor in the materials and methods part, 2-5. Analysis of mutation frequency as follows;

In this experimental condition, no change was observed in the shape of the fibroblasts under a light microscope.

Reviewer 2 Report

The work reported investigates pre-conditioning effects of different forms of low fluence irradiaiton. The work is of high quality and the Referee is happy to recommend its acceptance. There are a number of quite minor points that the Authors might like to consider prior to publication.

  1. The Authors claim the pretreatment dose is ultra low fluence, This is quite correct. It would be nice if these fluences were specified. Maybe a table with ion, fluence, flux, dose, LET and range in tissue would structure this information nicely.
  2. Was the  the time interval between the pretreatment dose and the  challenge dose standardised? If so what was the  interval.
  3. The equation on p. 3 line 122 is strictly dimensionally incorrect. As deposited dose is J/kg the density needs to be included. Most biological tissues have a density close to one so its a few % error.
  4. line 123.
    Was any check made to see if the irradiation temperature had any effect? The Referee does not think this additional stress is important - but different environments and time outside of the 37 °C incubator might have an effect.
  5. Line 134
    The Cross-stripe Irradiation method is confusing. If the Referee has understood correctly a He beam focused to 10 µm by  10 µm beam spot was rastered over 5625 300 µm x 300 µm square areas.
  6. Line specify "Spot size" and how this is defined.

Author Response

Our replies for the reviewer’s (2) comments

We appreciate for reviewing our manuscript and insightful comments and valuable suggestions. We revised the manuscript according to the suggestions and comments from the reviewers.

  1. The Authors claim the pretreatment dose is ultra low fluence, This is quite correct. It would be nice if these fluences were specified. Maybe a table with ion, fluence, flux, dose, LET and range in tissue would structure this information nicely.

We added characteristics, such as energy, LET, fluence at 0.1cGy and average number of hits/nucleus, of heavy ions that were used for pre-treatment in Table 1 as follows;

Table 1.Characteristics of pre-treatment of heavy-ion beams

Pre-treatment radiation types

Energy (MeV/n)

LET

(keV/µm)

Fluence at 0.1 cGy

(ions/cm2)

Average number of hits/nucleusa (%)

Helium ions

150

2.3

2.71 x 105

61

Carbon ions

290

13.3

4.69 x 104

15

Iron ions

500

200

3.12 x 103

1.1

aPercent of direct hit / cell nucleus were calculated according to the Poisson distribution using the measured diameter of NB1RGB cells (21 µm in diameter).

  1. Was the time interval between the pretreatment dose and the challenge dose standardised? If so what was the interval.

In this experiment the cells were first irradiated with the pre-treatment and then irradiated with the challenging dose of X rays. The physical reason, which was the time to move from the facilities of the pretreated irradiations to the facility of X-ray irradiation, needed for at least 30 min. So we chose 30 min as the time interval between the pretreatment and the challenge dose.

  1. The equation on p. 3 line 122 is strictly dimensionally incorrect. As deposited dose is J/kg the density needs to be included. Most biological tissues have a density close to one so its a few % error.

In the case of particle radiations, especially heavy ions, we calculated irradiation doses as (particle numbers) x (its LET value) x 1.602x10-9.

Now we assume the energy loss when single ion with [LET] (keV/µm) passes in a volume of 1µm x 1µm x1µm [V].

In this situation the energy deposition [E] in a volume [V]] is [E] = [LET] (keV/µm3).

Here 1eV is 1.602 x 10-19(J) and 1keV is 1.602 x 10-16(J).

And 1 m3is approximated as 1 x 103kg (a density close to one), then 1 µm3is 1x10-15kg.

          So [E] = [LET] x 1.602 x10-1(J/kg).

In the scintillation counter we usually measure the particle number / cm2.

So single ion / µm2is 1 x 108ions / cm2.

Here [E] (J/kg) is to be the absorption dose of 1 x 108ions. Therefore when single ion passes the area of cm2in the scintillation counter, the dose [e] is

[e] = [E] / 1 x 108 = [LET] x 1.602 x 10-9(J/kg).

Finally, we get the formula as follows;

  1. line 123. 
    Was any check made to see if the irradiation temperature had any effect? The Referee does not think this additional stress is important - but different environments and time outside of the 37 °C incubator might have an effect.

We agree with the Referee’s opinion. One of the important factors for cellular responses is temperature. In this study we unified all irradiation temperature at 22-24oC. We have no data what kinds of cellular responses were induced at 22-24oC and were not induced at 22-24oC. We believe this is an important research subject and a great task in the research field of radio-biological study. We would like propose further elucidate study, containing other pathway of bystander effect such as secreted factor(s), not gap-junction mediated cell-to-cell communication. We added a sentence in the discussion part as follows;

Altogether, our findings indicate that bystander effect induced by gap-junction mediated cell-to-cell communication is one possible modulator of radiation-induced epigenetic effects. However, the mechanism of radiation-induced bystander effects is still unknown and we need to study in more detail, e.g. what kinds of molecules bear the pathway of gap-junction mediated cell-to-cell communication. Also, other pathway, such as a secreted factor to culture medium from direct irradiated cells, is high-powered candidate for inducing bystander effect. Using healthy human cells we observed secreted-factor induced bystander effect[27]. To further elucidate this mechanism, researchers should explore which genes and proteins could be critical player on bystander effects.

  1. Line 134
    The Cross-stripe Irradiation method is confusing. If the Referee has understood correctly a He beam focused to 10 µm by 10 µm beam spot was rastered over 5625 300 µm x 300 µm square areas.
  2. Line specify "Spot size" and how this is defined.

The spot size “10µm x 10µm” means if 10 protons are irradiated by this spot size, 10 protons are hit in this area (100% hit). Our cross-stripe irradiation lines up 75 x 75 neighboring spots of 10µm x 10µm at the intervals of 300µm in all directions. We added the sentence in the materials and methods part, 2-3. Proton microbeam irradiation as follows;

The irradiation protocol was carried out using the -cross-stripe irradiation method (Figure 1). The spot size of the microbeam was 10µm10µm and irradiation spots line up 5625 (7575) neighboring spots at the intervals of 300µm in all directions. In this experimental setting, the total cell number in the confluent state was calculated to be cells/irradiation dish. Based on this calculated cell density, the rough estimated percentage of cells irradiated with a single proton, under a  microbeam size, was . The beam interval of the applied irradiation method was 300 µm, which is considerably bigger than the size of a single cell that was determined to be 21 µmin diameter of cell nuclei under the microscope.
